# Characterization of the Molecular Events Underlying the Establishment of Axillary Meristem Region in Pepper

**DOI:** 10.3390/ijms241612718

**Published:** 2023-08-12

**Authors:** Haoran Wang, Sujun Liu, Shijie Ma, Yun Wang, Hanyu Yang, Jiankun Liu, Mingxuan Li, Xiangyun Cui, Sun Liang, Qing Cheng, Huolin Shen

**Affiliations:** 1Department of Vegetable Science, College of Horticulture, China Agricultural University, Beijing 100193, China; 18813013760@163.com (H.W.); liusujun213@163.com (S.L.); msj15130269191@163.com (S.M.); cloudking1010@163.com (Y.W.); sy20223173025@cau.edu.cn (H.Y.); liujiankun0216@163.com (J.L.); q1115323473@Outlook.com (M.L.); c13513430025@163.com (X.C.); liang_sun@cau.edu.cn (S.L.); 2Beijing Key Laboratory of Growth and Developmental Regulation for Protected Vegetable Crops, China Agricultural University, Beijing 100193, China; 3Sanya Institute, China Agricultural University, Sanya 572025, China

**Keywords:** pepper, lateral branch, axillary meristem, comparative transcriptome, IAA, ABA, auxin transport

## Abstract

Plant architecture is a major motif of plant diversity, and shoot branching patterns primarily determine the aerial architecture of plants. In this study, we identified an inbred pepper line with fewer lateral branches, 20C1734, which was free of lateral branches at the middle and upper nodes of the main stem with smooth and flat leaf axils. Successive leaf axil sections confirmed that in normal pepper plants, for either node n, P_n_ (Primordium n) < 1 cm and P_n+1_ < 1 cm were the critical periods between the identification of axillary meristems and the establishment of the region, whereas P_n+3_ < 1 cm was fully developed and formed a completely new organ. In 20C1734, the normal axillary meristematic tissue region establishment and meristematic cell identity confirmation could not be performed on the axils without axillary buds. Comparative transcriptome analysis revealed that “auxin-activated signaling pathway”, “response to auxin”, “response to abscisic acid”, “auxin biosynthetic process”, and the biosynthesis of the terms/pathways, such as “secondary metabolites”, were differentially enriched in different types of leaf axils at critical periods of axillary meristem development. The accuracy of RNA-seq was verified using RT-PCR for some genes in the pathway. Several differentially expressed genes (DEGs) related to endogenous phytohormones were targeted, including several genes of the PINs family. The endogenous hormone assay showed extremely high levels of IAA and ABA in leaf axils without axillary buds. ABA content in particular was unusually high. At the same time, there is no regular change in IAA level in this type of leaf axils (normal leaf axils will be accompanied by AM formation and IAA content will be low). Based on this, we speculated that the contents of endogenous hormones IAA and ABA in 20C1734 plant increased sharply, which led to the abnormal expression of genes in related pathways, which affected the formation of Ams in leaf axils in the middle and late vegetative growth period, and finally, nodes without axillary buds and side branches appeared.

## 1. Introduction

The branch and inflorescence structures of flowering plants largely depend on meristem activity [1]. Stem apical meristems (SAMs) and root apical meristem are two groups of pluripotent stem cells completes growth and development by continuously increasing the number of plant nodes (called phytomeres) as growth units [2,3]. Each phytomere normally consist of three positions: stem segments of internodes, leaves (developed from the leaf primordium), and axillary buds/lateral branches in the axils of leaves (developed from the axillary meristem (AM)) [4,5]. The activity and quantity of AMs largely determine plant strain and adaptability [6].

Lateral branching is an important agronomic trait in crops that directly determines plant architecture [7] and has a very important effect on light capture, photosynthesis, and resource allocation in plants, thereby changing the yield and quality of crops [8]. As described above, lateral branches develop from AMs in the leaf axils. These meristems grow and develop into axillary buds that remain dormant during the nutritional growth phase or continue to grow to form branches [9]. It is generally believed that the formation of lateral branches involves two processes: the initiation of lateral buds (formation and activation of AMs) and the germination of axillary buds to become lateral branches [10]. Regarding the first process, extensive molecular genetic studies have revealed a conserved *LATERAL SUPPRESSOR*/*LATERAL SUPPRESSOR*/*MONOCULM1* (*LS*/*LAS*/*MOC1*) genetic pathway controlling the initiation of AM in both dicots and monocots; it is also one of the most important and well-researched pathways [11]. *MOC1* was identified as a regulator of tiller formation in rice. The functional deletion mutant *moc1* has only one main stem without tillers [12], and mutants of homologous genes in tomato [13] and *Arabidopsis* [14] have similar phenotypes. Another tiller mutant in rice, *moc3*, showed a phenotype similar to that of *moc1*, in which AMs were rapidly inhibited, and normal axillary buds could not be formed [15]. In addition, *LAX1* [16,17] and *LAX2* [18] in rice; *REV* [19,20], *ROX* [14], and *RAX1-3* [21] in *Arabidopsis*; and *BL*/*TO* [22] in tomato were all verified to regulate AM formation in different ways, and their mutants/edited materials all showed axillary bud reduction to different degrees. In summary, auxin moves down from the shoot apex inhibiting while strigolactone (undiscovered at the time) moves up from the roots and inhibits through the other genes [23].

Phytohormones also play important roles in the formation and development of lateral buds/branches, including auxin (IAA), cytokinin (CK), and abscisic acid (ABA), and numerous molecular biology studies have been performed around them [24]. One of the best-known functions of auxin for lateral branches is to inhibit axillary bud growth via the apical dominance pathway [25], where in fact auxin participates in two phases of lateral branch formation. During vegetative development, leaves are initiated in a regular patterns from cells at the flanks of the SAM. Transport of the mobile phytohormone auxin in the epidermal layer (L1 layer) of the SAM leads to the formation of auxin maxima, which determines the positions of leaf initiation [26]. After this, it is necessary to rapidly efflux auxin to ensure the minimum concentration of auxin to realize the regional identification and establishment of AMs [27,28], and the boundary regions between the SAM and leaf primordium show first high and then low auxin concentrations at an early stage before AM initiation, which was demonstrated using the auxin indicators *DII* and *DR5* [29]. From this, it can be found that auxin transport and concentration maintenance play an important role. In *Arabidopsis*, the major auxin influx carriers are *AUXIN RESISTANT1* (*AUX1*), *LIKE-AUX1* (*LAX1*), and *LAX2* [30], whereas the main auxin efflux carrier is *PIN* gene family [31]. From this, it can be concluded that auxin transport inhibitor NPA will lead to the reduction of axillary buds and the emergence of empty leaf axils in *Arabidopsis*, which cause the block in polar auxin transport, leading to a defect in AMs formation [32]. In the absence of *PIN1* activity, *Arabidopsis* meristems form only few lateral organs, resulting in a pin-like stem architecture [33]. Studies on rice show that *RFL* can positively regulate auxin transport in stems, thus controlling specification and outgrowth of vegetative AMs [34].

As an antagonist of auxin, Cytokinins (CKs) also play a role in two stages of collateral development [35]. The phenotype of tomato *Bl* allele *TOROSA-2* mutant (*to-2*) is related to the low CK levels, and exogenous application can cause AMs to start [36], and a follow-up study found that initiation of AMs requires a cytokinin signaling pulse [37]. A subsequent study found that initiation in the *supershoot* mutants of *Arabidopsis*, whose endogenous CK level is 3–9 times higher than that of wild type; this leads to the formation of multiple AMs in the axils of rosette leaves and stem leaves [38]. ABA is well known for its role in plant adaptation to abiotic stress [39]. In the regulation of axillary buds, researchers first found that its exogenous supply can inhibit bud growth [40]. Subsequent studies have found that ABA can partially inhibit the growth of buds by reducing the biosynthesis and transport of auxin in buds and cell proliferation [41]; this may damage the ability of buds to export their own auxin and grow [42]. There are limited reports on the role of ABA in axillary bud formation. Earlier studies found that rice *TILLER ENHANCER* (*TE*) encodes the activator of APC/C^TE^ E3 ubiquitin ligase complex, and *MOC1* is regulated by ABA content to inhibit tillering [43]. The determination of endogenous hormones in tomato unbranched ls mutant showed that the contents of IAA and ABA increased sharply [44]. Recent studies have found that a high level of GAs can activate the complex, thus promoting the degradation of MOC1 and reducing tillering, while a slightly higher level of ABA can maintain the tillering situation in the middle and upper parts of the ground by antagonizing GAs [45]. In the same year, the endogenous ABA content in the high tillering mutant *t20* found in rice decreased, and ABA-deficient mutants *aba1* and *aba2* had more unproductive upper tillers at maturity [46]. On the whole, it is widely verified that the influence of ABA on AMs initiation mainly acts on the middle and late stages of vegetative growth, which may be related to the accumulation period of ABA in plants [47]. 

Although current research focuses on phytohormones, especially on the role of auxin in axillary shoot growth, relatively little is known about the regulatory effects of auxin on AM formation, and they have been described only rarely in pepper plants. Pepper (*Capsicum annuum*) is an important vegetable crop, and its growth pattern differs from that of rice and *Arabidopsis* [48]. In this study, we conducted cytological observations on the pepper variety 20C1734 with reduced lateral branched and variety 20C1733 with the normal lateral branch pattern. First, we determined each pepper AM development process’s specific time and judgment standards. The experiment revealed that AMs did not form at nodes without axillary buds in 20C1734, the absence of meristematic cells in the leaf axil region, and also the presence of abnormalities from the establishment of the AM region. Based on the cytological results, we finally sampled the leaf axillary areas on different materials with and without axillary buds and retouched them under a microscope. A comparative transcriptome analysis using RNA-seq locked the “auxin-activated signaling pathway”, “response to auxin”, and “plant hormone signal transduction” as enrichment terms/pathways during the critical period of AM formation. Endogenous phytohormone content measurements in the same samples demonstrated that differences in auxin and abscisic acid levels were potential factors contributing to the failure of AM formation. Auxin transport may be a critical factor, which is corroborated by the abnormal expression of related genes detected using RNA-seq and RT-PCR.

## 2. Results

### 2.1. Phenotypic Observation of Pepper 20C1733 and 20C1734 and AMs Development Stage of Pepper

Pepper plants have falsely dichotomous branching structure. Except for a few strains with determinate growth, the lateral branches below the point of initial flower expansion are not conducive to the production of peppers but consume excessive nutrients and waste labor costs for pruning, which is not conducive to light simplified cultivation [49,50]. An inbred pepper inbred line with reduced lateral branches (20C1734) was screened in our laboratory in a previous field experiment and compared with the normal lateral branch inbred line 20C1733. We found that the epidermis of 20C1734 had a velvet appearance and had more nodes than 20C1733 (18–21). Both are annual pepper species; however, the fruit types are Alstonia pepper and line pepper, respectively, and they are not directly related. To illustrate the situation at each node more clearly, we removed the leaves below the swelling point at the top of the materials (Figure 1a). During the growth stages, the lateral branches decreased significantly, and there were lateral branches in the axils of the bottom 4–5 true leaves (Figure 1a,c). There were no lateral branches/axillary buds in the upper node, and the connection between the stem and petiole was smooth and flat and remained unchanged until death (Figure 1a,d). In comparison, the number of nodes of 20C1733 was less (10–13), and lateral branches existed at each node in the vegetative growth stage (Figure 1a,b). By counting the number of lateral branches of 12 plants planted in the growth room, as shown in Appendix A, it was found that the proportion of lateral branches of 20C1733 was 100%; that is, there were lateral branches at each node. However, only the first five nodes of 20C1734 had collateral branches, the proportion of collateral branches ranged from 11.11% to 27.78%, and the subsequent nodes had a stable collateral loss. In addition, the development of flowering, fruit set, and reproductive lateral branches were unaffected in 20C1734 (Figure 1a).

In studying the presence/absence of axillary buds, it is important to examine the timing of AM formation during development. To clarify this point, we first observed the AM development process at the axil of the sixth node of the normal lateral branch material 20C1733 and determined the specific stages of AM development in pepper. Based on the research process of tomato and *Arabidopsis*, using the development degree of the leaf primordium as the time criterion, the axils of P_6_ leaves with P_6_ (Primordium 6) (the sixth true leaf, similar in the follow-up) < 1 cm, P_6_ > 1 cm but P_7_ < 1 cm, P_8_ < 1 cm, P_9_ < 1 cm, and P_10_ < 1 cm were selected, and the organs, including the stem, leaf primordium/petiole, and their connections, were cut out to be used for embedding, slicing, and dyeing. By observing and taking pictures under a microscope, we found that at P_6_ < 1 cm, the leaf axil showed a conventional cell arrangement, the cells below the epidermal layer were uniform in size, and no tissue formation was observed (Figure 1e). When P_6_ > 1 cm but P_7_ < 1 cm, the leaf axil bulged slightly; numerous regenerative cells with small size, invisible cell walls, and concentrated rows toward the tip appeared below the epidermal layer at the intersection angle; and the AM tissue region was established (Figure 1f). When P_8_ < 1 cm, there was a significant bulge in the leaf axil, the number of cells increased further, and the AM tissue was initially formed (Figure 1g). For P_9_ < 1 cm, we found that the AM in the leaf axil of P_6_ was completely formed and gradually became larger with time, developing into a complete AM tissue with a well-formed internal structure that could be observed and distinguished by identification (Figure 1h), until finally axillary buds were formed at P_6_ (Figure 1i).

### 2.2. Morphology of the 20C1733 and 20C1734 Leaf Axils

From the above results, it can be seen that when the leaf primordium n (P_n_) < 1 cm, the AM in the leaf axilla has not been produced; when the P_n+1_ < 1 cm, the cells in the leaf axils were transformed into differentiated cells, which were in the stage of AM region establishment; or when the P_n+3_ < 1 cm, the AM in the leaf axilla of P_n_ was completely formed. We divide it into three AM development stages (1, 2 and 3) in this study, corresponding to Figure 1e,f,h, respectively, which will be described in detail later. Based on the similarities and differences in plant materials, we selected different stages of leaf axils at P_3_ (Figure 1c, with axillary buds) and P_9_ (Figure 1d, without axillary buds) of 20C1734 and the leaf axils at P_9_ (Figure 1b, with axillary buds) of 20C1733 for observation and comparison.

In stage 1, whether node P_3_ had axillary buds in 20C1734 (Figure 2a,d), the node P_9_ without axillary buds in 20C1733 (Figure 2b,e) or the nodal position P_9_ with axillary buds in 20C1733 (Figure 2c,f), the cells in the leaf axils were evenly distributed, uniform in size, with thick cell walls, typical traits, and no AM tissue formation, which were differentiated cells. The difference between the positions appeared in stage 2. The AM tissue area in the P_9_ leaf axil of 20C1733 was established (Figure 2i,l), resulting in a small bulge and dense cell structure. The cells beneath the epidermal layer started to dedifferentiate, becoming smaller and irregularly polyhedral, with thin cell walls and clearly visible nuclei. Meanwhile, P_3_ (Figure 2g,j) and P_9_ (Figure 2h,k) of 20C1734 did not change, and no new tissue was formed. By stage 3, AM tissue in the P_9_ leaf axil of 20C1733 was completely formed (Figure 2o,r), forming a complete bulge and organ structure and maintaining a dense area of meristematic cells at the center of the nascent organ. In contrast, the P_3_ leaf axil of 20C1734 started to form a bulge (Figure 2m,p), whereas the P_9_ leaf axil remained in its original state with loose and mature cells and no meristematic cells or AM formation (Figure 2n,q).

### 2.3. Overview of the RNA-Seq Data

After the differences between the leaf axils with and without axillary buds were determined using cytological tests, we performed a comparative analysis using RNA-seq. The three positions of the two materials in Figure 2 are labeled A, B, and C. Combined with the observation results in Figure 1, three stages (1, 2, and 3) were sampled at each position; thus, nine different developmental stages and types of leaf axils were sequenced and analyzed (Table 1 and Appendix A). For the nine categories of samples tested, three biological replicates were collected and marked, for example, A1-1, A1-2, A1-3, and A2-1. The detected reads ranged from 19,300,909–26,023,912, with the lowest Q30 reaching 90.90% (Appendix A), indicating that the RNA-seq data met the requirements for subsequent analyses.

In this study, DEGs were identified using a threshold *p*-value of ≤0.05 and an absolute value of log_2_Ratio ≥ 1, and Ca_59 was selected as the reference genome for analysis. We conducted a comparative transcriptome analysis to better understand the developmental process of AM and the development of leaf axils with or without axillary buds. As shown in Figure 3a, in the leaf axils with axillary buds in 20C1734, a large number of DEGs (2249) were enriched between A2 and A3, which was much larger than that between A1 and A2 (558). We observed the same expression trend for position B (Figure 3b); however, the number of DEGs at position B was lower than that at position A during the same developmental stage. Figure 3c shows position C with axillary buds, which shows gene expression during normal AM development. The number of DEGs between C1 and C2 (2653) were higher than that between C2 and C3 (1612), which was different than the trends observed for positions A and B. Combined with the developmental process, we concluded that the process of establishment of the AM region is the main enrichment time for DEGs (A2 vs. A3, C1 vs. C2). Throughout the process of AM development, the DEGs detected at positions A and C were almost the same, which was much higher than that at position B. This indicated that position B was indeed abnormal in the development of AM, resulting in the silencing of genes whose expression levels should have changed.

Subsequently, we compared the developmental stages of the different positions. As shown in Figure 3d, we compared the nodes with (A) and without (B) axillary bud in 20C1734. The number of DEGs between the two regions were relatively stable in stages 1 and 2, and there was no significant difference in the number between different periods. However, in the third stage, the number of DEGs between A3 and B3 increased significantly, reaching 3215, which is also consistent with our cytology results (Figure 2m,n), at which point the identification and area establishment of AM occurred at position A. Additionally, the leaf axils of 20C1733 with axillary buds (C) and of 20C1734 without axillary buds (B) were compared; they were all in the P_9_ position. The number of DEGs of stages 2 and 3 were very similar (approximately 2600) while many DEGs were mainly concentrated in stage 1. The DEGs of C1 vs. B1 reached 4624, which was almost equal to the sum of the other two stages. This suggests that there are a certain number of gene expression changes from before AM formation to AM region establishment and identity confirmation, which may be a prerequisite for the establishment of the AM region. All DEGs obtained for the comparison in (Figure 3) are listed in Appendix A.

### 2.4. GO and KEGG Analysis of Comparative Transcriptomes

We identified the critical period by analyzing the DEGs between different samples: the AM area establishment process. Based on this, we analyzed the GO annotations and KEGG pathways. First, we compared B1 with B2 and C1 with C2. At this critical stage of AM development, B1 vs. B2 accumulated fewer GO terms, and fewer genes were enriched in each term than that for C1 vs. C2 (Appendix A). In terms of biological processes (BPs), the main enriched terms were “response to abscisic acid”, “response to wounding”, and “response to oxidative stress” (Figure 4a,b). However, it is worth noting that the “auxin-activated signaling pathway” was additionally enriched in position C. In the cellular component (CC) subgroup, they were almost identical, and the “integral component of membrane” was the most abundant term. In the molecular function (MF) category, “transcription factor activity” and “oxidoreductase activity” were significantly enriched. Similarly, in the KEGG analysis of this period, position B showed lower enrichment than the other two positions (Figure 4c,d, Appendix A), and the “biosynthesis of secondary metabolites,” “metabolic pathways”, and “galactose metabolism” pathways were enriched in both comparison groups. However, some additional genes were enriched in the pathways of “plant hormone signal transduction” and “carotenoid biosynthesis” in position C. It was found that carotenoid synthesis was influenced by plant hormones [51]. Based on this and the results of GO and KEGG analyses, we tentatively concluded that plant hormones, signaling had a certain regulatory effect on the establishment of the AM domain.

Next, we compared B3 and B2 and C3 and C2. The overall trend in this phase was the same as in the previous paragraph (Appendix A); terms and DEGs of each term in position B were still less than those in C. The difference mainly occurred in the BP category, “regulation of transcription, DNA-templated” was the most significantly enriched term (Figure 5a,b), and the two terms “response to auxin” and “response to jasmonic acid” were enriched only in position C, which is consistent with our conclusion in the previous paragraph. In the KEGG analysis, “biosynthesis of secondary metabolites” was the most enriched pathway (Figure 5c,d), and “phenylpropanoid biosynthesis” and “biosynthesis of various plant secondary metabolites” were also present in both comparison groups. Similarly, at this stage, “plant hormone signal transduction” was still present only in the comparator group of position C, and numerous genes in this position were enriched in “metabolic pathways.” Based on the above comparison results, we found that compared with position C, position B lacked enrichment of the related term/pathway of hormone response and transport during the entire detection stage, and the silencing of these genes may have caused the abnormal deletion of the AM region of 20C1734.

Finally, we analyzed the enrichment of positions B and C at the same developmental stage. As shown in Figure 6a, the main enrichment of the two subgroups CC and MF at stage 1 was similar to the previous two segments, and “integral component of membrane” and “transcription factor/protein (enzyme)/ion activity and binding properties” were the main enrichment terms, respectively. In terms of BP, the “response to abscisic acid”, “cell differentiation”, and “auxin biosynthetic process” were still differentially enriched terms, and the number of DEGs was higher and more significant. In stage 2 (Figure 6b), the terms enriched in the CC and MF subgroups remained essentially unchanged. In the BP subgroup, “response to abscisic acid” was still present while the number of DEGs enriched in “auxin biosynthetic process” in the previous stage decreased but was still detected. In contrast, the number of DEGs in “response to auxin” and “response to jasmonic acid” increased to become the significantly enriched terms in this stage (Appendix A). This became more evident when the KEGG was analyzed. In addition to “metabolic pathways” and “biosynthesis of secondary metabolites”, which were always present, “plant hormone signal transduction” was significantly enriched between stages, showing the third highest number of DEGs. The GO and KEGG analyses of A2, B2, A3, and B3 are shown in Appendix A, from which consistent results were obtained. The GO terms and KEGG pathways obtained for all comparison groups are summarized in Appendix A.

By analyzing the GO and KEGG enrichment results between the different comparison groups, after removing the DEGs-rich categories and pathways that co-exist between nodes with and without collateral branches in different comparison groups, we found that the differential pathways of auxin and abscisic acid run through the whole process of AMs development. This led to the expression changes of pathway genes, such as “cell differentiation” and “cell wall”. Therefore, we suggest that AM can establish normal areas and even form definite and developable tissues later; this process is regulated by auxin and abscisic acid, and in these processes, the transportation, signaling and transformation of auxin, and the reaction caused by abscisic acid content are more important.

### 2.5. Expression Verification of Partial DEGs in Enrichment Pathway

Through comparative analysis of transcriptome data, we identified several auxin-related enrichment pathways that may be involved in the regional establishment and confirmation of AM identity. For this, we extracted the sequencing results of genes enriched in these pathways from the C1 vs. B1 comparison group and generated a heat map. We found that most genes of the term “auxin-activated signaling pathway” (Figure 7a above the black line) and the “plant hormone signal transduction” pathway (Figure 7a below which) were significantly expressed at position C at this stage. Several genes were selected for RT-PCR analysis to verify the accuracy of the RNA-seq data and further explore related genes. The auxin-related genes that were not in the significantly enriched pathway included *PIN3*, *PIN4*, and *PIN6* of the auxin transporter family; the auxin response factors *ARF1* and *ARF6*; and the auxin-responsive protein *IAA27* (Figure 7b). The transcriptome data of the same genes are shown in Figure 7c. The difference in the expression of the same gene between them was consistent, and the degree of differential expression of individual genes was slightly different, with several genes of the PIN transporter family showing lower expression at position B, while *ARF6* and *IAA27* showed lower expression at position C. RT-PCR analyses of *ARF1* and *ARF6* revealed no differences in the expression of these genes. Figure 7d shows the RT-PCR results of several genes in the “plant hormone signal transduction” pathway, and Figure 7e shows their transcriptome detection results. The expression trends of these genes were also consistent with the RNA-seq results. Among them, *ALDO3* (a plant hormone precursor) was enriched at position B, and the remaining hormone transport-related genes were more enriched at position C.

### 2.6. Determination of IAA, TZR, GA_3_, and ABA in Different Stages and Positions

Combining the distribution of DEGs and the results of the GO and KEGG pathway analyses, we correlated the AM region establishment process with auxin levels, which was verified using RT-PCR (Figure 7); at the same time, abscisic acid also continuously participated in the process of establishing AM region. Next, we prepared samples of the same standard as the transcriptome sequencing (Table 1) and measured and analyzed the changes in the endogenous hormones auxin (IAA), trans-zeatin-riboside (TZR), Gibberellin A3 (GA_3_), and abscisic acid (ABA). The IAA content at position B was significantly higher than at position C at the same node, whereas the content at position A was the lowest (Figure 8a). The red arrows marked at position A and C in the figure indicate the establishment stage of the AM region confirmed by cytological observation and transcriptome comparison. The IAA content generally decreased before AM formation, and the lowest IAA content was detected during AM region establishment (Figure 8a, position A, stage 3; position C, stage 2) and the release of previously accumulated IAA. After AM formation, the IAA content must increase to promote growth (Figure 8a, position C, stage 3). However, we did not observe such a pattern at position B, where the IAA content remained extremely high, and there was no regular change. 

The trend of TZR content (Figure 8b) was similar to that of IAA but showed continuous accumulation at position A and C during different time periods, in contrast to Figure 8a, which is inconsistent with our conventional understanding of the antagonistic effects of IAA and TZR. No regular changes that could be associated with the establishment of the AM region were observed. Figure 8c shows the content of GA_3_. Positions A and C generally showed the same trend as IAA, that is, low levels at the beginning of the AM region establishment. However, the degree of abnormal content in position B was much lower than that of IAA, and the difference was insignificant. What surprised us most was the ABA content. As shown in Figure 8d, the ABA content in parts A and C where AMs existed was similar, and there was no regular change trend with the development of organs. However, in the part B without AMs, the ABA content was surprisingly increased, reaching more than ten times that of the part C, which was rare in the process of plant vegetative development, and the high level of this lasted for the whole cycle in the leaf axils.

Interestingly, among the four hormones examined in position B, although the magnitude of the change in IAA and ABA were the most significant, the levels of TZR and GA_3_ were almost always higher in position B than in positions A and C at the same stage, although the differences were not significant. Based on this, we propose that hormone homeostasis at this site may be disturbed, resulting in a general increase in hormone levels at position B; however, IAA and ABA may play the most significant role.

### 2.7. DR5::Ruby Instantly Transforms Pepper

For the high expression of auxin, we speculate that there are two possibilities. The first point is that auxin synthesis genes are significantly expressed in the leaf axil position of axil-less buds of 20C1734, but RNA-seq assays revealed no abnormal changes in the expression of auxin synthesis genes *TAR1*/*TAR2* and *YUC* family genes during the critical period of AM formation (Appendix A). The second point is that auxin transport is abnormal at this location, where partial auxin transport/signal transduction-related DEGs were identified in a previous comparative transcriptome description, and their expression changes during the critical period provide ideas for our subsequent studies. In order to find out the role of auxin in regulating axillary bud production, we infected the less lateral branch material 20C1734 by vacuumizing, which was proven to be feasible in strawberry. Based on this, we selected the auxin reporter gene *DIRECT REPEAT5* (*DR5*), a synthetic promoter constructs that acts as a readout for auxin activity. We constructed the *DR5::Ruby* vector, transformed it into *Agrobacterium*, and then prepared the infestation solution and infested it.

Figure 9a shows the comparison of *DR5::Ruby* plants (left) and non-infected plants (right) after vacuum infection for 25 days, in which purple–red traces can be observed visually at the stem, part of the petiole and the leaf blade of DR5::Ruby plants, with the most obvious and extensive display in the stem, consistent with the result that it serves as the main channel for auxin transport. This phenomenon was more clearly visualized when the petiole was retained and the leaf blade was removed (Figure 9b). The red line in Figure 9c serves as a dividing line to distinguish the two types of leaf axils. The upper part is a leaf axil without axillary buds, and the lower part is a leaf axil with axillary buds. As shown in Figure 9d, at the leaf axils with axillary buds, the purplish red color runs through the joint of stem and petiole at the node, and continues to spread forward to the base of axillary bud. In contrast, at the axils of leaves without axillary buds, the purplish red color is observed only on the stem, did not transition to the petiole base, and there was no auxin reaction at the base of leaf axils. (Figure 9e). Furthermore, we photographed the front and side of each leaf axils under the stereomicroscope, as shown in Figure 9f. The leaf axils with axillary buds to the left of the red line were able to observe a purple-red Ruby spreading to the base of the axillary buds from P_1_ to P_5_, marked by blue lines, it can be found that Ruby’s reactions have crossed the position of axillary buds. While similar phenotypes could not be observed for each node without axillary buds on the right side, from P_6_ to P_15_. Based on the red line, we can find that there is no transition to the angle between the main stem and the petiole. To demonstrate this more visually, we drew a schematic diagram as shown on the left side of Figure 9g. The red line is used as a marker to distinguish the crease where the stem is attached to the leaf. We can clearly see that *DR5::Ruby* spreads across the red line at the leaf axils with axillary buds in the upper panel and continues to the base of the axillary bud while at the leaf axils without axillary buds in the lower panel, *DR5::Ruby* stops abruptly at the boundary, as observed visually in Figure 9e,f, indicating that the auxin response may not be possible here.

## 3. Discussion

### 3.1. The AM of 20C1734 Is Abnormal

Shoot branching is an important trait that affects crop production [52]. The ability of axillary buds to grow into lateral branches depends on the growth force of the axillary buds. In contrast, the presence/absence of axillary buds depends on the ability of the AM to complete the normal formation of regional establishment [53]. The inbred pepper line 20C1734 was smooth at the nodes in the middle and late stage of nutritional development, with no axillary buds and no abnormal flowering and fruiting (Figure 1a). For false dichotomous branching peppers, most of these nutritional stages of branching are pointless, require labor costs for shaping and tending, and disperse nutrients that are not beneficial to plant growth. Similarly, in rice, it is believed that although more panicle branches have better potential to increase yield, plants with this trait tend to have a low seed setting percentage because nutrition is limited [49].

AM production is dependent on and occurs later than in the leaf primordium in which it is located [54]. During the nutritional growth stage in *Arabidopsis*, the initiation of AMs starts in the axils of the oldest leaves and then progresses toward younger leaf axils, resulting in an acropetal gradient of lateral shoot formation [55,56], similar to that in peppers. During prolonged vegetative development in the *Arabidopsis* accession Col-0, AM formation was first detected using the meristem marker *STM* in the 16th leaf axil (P_16_), counted from the SAM and as a morphological structure in the axils of the P_21_/P_22_ primordium [57]. However, another study suggested that AM initiation in *Arabidopsis* occurred much earlier, with the *STM* signal detected at the P_11_ stage, and the subsequent establishment of the regional identity of AM was observed more rapidly in morphology [58]. In tomatoes, when the leaf primordium closest to the SAM is P_1_, AM formation manifests morphologically as bulges at the P_6_–P_7_ leaf axils [59], and a similar pattern was found in sunflowers [60]. Identifying the various stages of AM development is necessary for follow-up studies, and we used the leaf primordium base as a reference. Considering the longer nutritional growth cycle of pepper, we numbered the true leaves sequentially, with the leaf primordium of the first true leaf being P_1_ and upwards. The normal lateral branch material 20C1733 was selected and sampled in series according to the developmental process (Appendix A). For any leaf primordium P_n_ < 1 cm, the leaf axils to which it belonged were normal cellular structures, and no tissue appeared (Figure 1e). When P_n+1_ < 1 cm, there was an inconspicuously raised new ground tissue area established with a large enrichment of cells, which were smaller in volume than their neighbors (Figure 1f) and presumably meristematic at this stage [61]. Until the stage of P_n+3_ < 1 cm, the AM at the axil of the leaf where P_n_ was located was significantly raised, forming a clear tissue with a dense arrangement of new cells (Figure 1h), and axillary buds were still not observed morphologically (Appendix A).

After understanding the various developmental stages of the AM in pepper, we examined the difference between 20C1734 with and without axillary bud nodes from a cytological perspective. For the axillary bud node P_3_, although the AM could form normally, the developmental time was later (two leaf primordium growths behind) than that of the normal strain (Figure 2g,m). In contrast, no changes in cell arrangement or type were detected from the beginning in the axillary budless node P_9_, and no bulge was found (Figure 2h,n), which is considered a precursor of AM formation in most plants [62]. In addition, it can be clearly seen in Figure 2k,q that the leaf axils without axillary buds are composed of differentiated cells and some of them have thicker cell walls, which are significantly different from those with axillary buds (Figure 2l,r). This is consistent with previous studies in *Arabidopsis* [63]. Two possible theories have been proposed for the formation of the AM: (1) it is generated by the de novo occurrence of new meristematic cells, or (2) it results from the detachment of existing meristem cells from the SAM [64]. Based on the cytological observations, we proposed that AM formation in peppers is the primary cause. On the one hand, developmental continuity between AM progenitor cells and the SAM is not readily apparent. In addition, leaf axils with/without axillary buds did not differ in either cell arrangement, number, or type before AM formation (Figure 2a–c). However, in the subsequent transition stage (Figure 2, g→m, c→i), cells at leaf axils redifferentiated into meristematic cells, increasing in number while thinning the cell wall and proliferating rapidly in the L2–L5 layer below the epidermis; in previous studies, this condition was considered as the characteristic of meristem cells [65]. Therefore, we propose that the identity of AM at the leaf axils was established independently of SAM, which was similarly verified during AM development in rice [66] and *Arabidopsis* [67].

### 3.2. Comparative Transcriptome Analysis Revealed the Enrichment of DEGs

A comparative transcriptome analysis of tissues at different developmental stages can provide valuable information on how the regulatory gene network controls specific development processes [68]. We identified the critical period as the AM region establishment phase by comparing the DEGs between different samples (Figure 3a–e), combined with cytological observations (Figure 2b,c,h,i). By comparing different samples, we found that “auxin-activated signaling pathway”, “response to auxin”, “response to jasmonic acid”, “response to abscisic acid”, and “auxin biosynthetic process” were the differentially expressed terms prevalent in position C in GO analysis. These terms were not found in position B and were more pronounced in C1 vs. B1, and C2 vs. B2 were more pronounced (Figure 6a,b). The KEGG analysis was more consistent, and the additional significant and continuously enriched “plant hormone signal transduction” pathway was rich in a large number of DEGs in addition to the two general pathways necessary for plant survival and growth, “metabolic pathways” and “biosynthesis of secondary metabolites” (Figure 6c,d). From this, we proposed that hormonal signaling, especially auxin and transport, are decisive in the establishment and formation of the AM region. A similar phenomenon was found in the rest of the plants.

The formation of lateral branches from AMs IS regulated by phytohormones such as CK, which act as positive regulators of branching, and auxin and SLs, which act as inhibitors of branching [69]. In *slb1* mutants of the woody plant *Liriodendron chinense*, key plant hormone signaling pathways involved in shoot branching may be deregulated, resulting in a multilateral branching phenotype [70]. Interestingly, during axillary shoot differentiation and regeneration in tissue culture, RNA-seq revealed that phytohormone signaling is the most important pathway [71]. In a transcriptome study of poplars, genes related to auxins or hormones that affect auxin signals were targets for optimizing branching [72]. An analysis of the interaction between *spi1* and genes regulating auxin transport indicates that auxin transport and biosynthesis synergistically regulate the formation of AMs and lateral organs in maize [73]. In addition to omics analyses, studies on several genes related to the regulation of AM formation have found that these genes are involved in phytohormone transport/signaling. For example, in rice, *RFL*, a gene that regulates AM formation, is associated with auxin [34], *OsVIL2* is involved in the strigolactone signaling pathway as a chromatin-interacting factor [74], and APC/C^TE^ E3 ubiquitin ligase interferes with tillering by coordinating the balance between abscisic acid and gibberellin [45]. *WUS* is activated by cytokinin signaling in *Arabidopsis* [75], and the deletion of the *ROXL* gene in sunflowers results in abnormal AM development. In contrast, while some genes of the auxin transport pathway are aberrantly expressed in the mutant [76].

Although it is also one of the more widespread terms, the analysis of the comparative transcriptome showed that “response to abscisic acid” appeared even more frequently than auxin-related. However, it was not unique to position C in each comparative group, but it was also present in B1 vs. B2 (Figure 4a,b) and B3 vs. B2 (Figure 5a,b). Therefore, we presume that, compared to auxin, the role of abscisic acid is not the main reason. On the one hand, the existence of this pathway runs through the samples of various periods/types, which is not enough to prove its correlation with AM development; at the same time, the “response to abscisic acid” is rich in a large number of stress-related genes corresponding to abscisic acid, which is in contrast to the change of auxin itself, so we do not think it is a potential regulatory factor. In this regard, we collated the expression of some genes in the “auxin-activated signaling pathway” and “plant hormone signal transduction” and generated a heat map (Figure 7a) and then proved the accuracy of RNA-seq using RT-PCR (Figure 7b,c). The validation results showed that the expression of several genes of the *PIN* auxin transporter family was significantly lower at position B. In contrast, the expression levels of auxin response factors, such as *ARF* and *IAA27*, at this site were higher than those at position C (Figure 7b), presumably due to an abnormal transit of auxin. Similarly, mutations in the growth hormone transporter proteins PIN1 and PID in *Arabidopsis* led to defects in AMs at the initiation stage, producing an abnormal plant architecture [77]. In maize, BIF1 and BIF4 interact with the activating auxin response factors (ARFs) and appear to regulate axillary meristems placement [78]. Similarly, in maize development, auxin biosynthesis, transport, and signaling controlled AM initiation and formation directly [79]. It has been reported that *LAX2* interacts with *LAX1*, which is involved with the auxin and brassinosteroid signal transduction pathways, to regulate the process of AM formation [80]. Previous studies have shown that auxins inhibit the maintenance of stem cells, which later give rise to AMs [37]. Altering auxin distribution or polar auxin transport using an auxin transport inhibitor or an auxin transport/signaling mutant inhibits the initiation of AMs, thereby interfering with shoot architecture [81,82].

### 3.3. The Formation of AM Is Related to Endogenous IAA and ABA

In a previous study, we found that auxin and abscisic acid has a potential regulatory role in the production of this trait. Hormonal assays revealed that endogenous auxin and abscisic acid levels were extremely high in 20C1734 leaf axils without axillary buds (Figure 8a). In addition, we found that for leaf axils where AM can be formed normally, a high-low-high auxin trend could be observed for the control cytology time node, with the lowest endogenous auxin content at the establishment of the AM region (Figure 8a, position C), whereas this decrease and change in trend could not be observed in position B without AM. However, abscisic acid showed a continuous and irregular high content. Previous studies have found that branches normally initiate as AMs at the site of auxin minima that form in the crease between newly emerging leaf primordium and the meristem [83]; in the boundary, the PIN1-dependent auxin minimum was also shown to promote AM formation, whereas ectopic auxin production in the boundary inhibits AM formation [84]. Auxin signaling in maize directly controls boundary domains during AMs formation [85], and an artificial increase in auxin in the developing boundary zone by localized expression of the auxin biosynthesis gene *iaaM* in transgenic *Arabidopsis* results in a lack of AMs in a portion of the leaf axils [86]. However, this minimum is not always the case, as the period of leaf primordium establishment is the maximum auxin in the leaf axil site and decreases thereafter, owing to the role of auxin transport [87].

The disruption of polar auxin transport compromises auxin depletion from the leaf axis and AMs initiation [88,89]. Mutations that disrupt auxin transport, signaling, and organ formation also disrupt AM formation, indicating that auxin or its transport is required [90]. The transient infestation of 20C1734 with *DR5::Ruby* showed that the purple–red Ruby spreading to the base of the axillary bud was observed at the nodes with axillary buds (Figure 9d,f) while at the nodes without axillary buds, *Ruby* spread from the stem and stopped at the crease between the stem and leaf primordium and did not spread to the leaf axils (Figure 9e–g), indicating that auxin transport was abnormal here, and the auxin accumulated at this site since the formation of leaf primordium could not be transported out. In *Arabidopsis* and tomato, AM initiation is characterized by preparative auxin depletion and subsequent meristem emergence via local auxin accumulation [91]. The *MAX1* gene (a specific repressor of nutritional axillary buds) in apples interferes with meristematic tissue formation by participating in auxin-polar transport [92]. In maize, the *BA1* gene, a rice *LAX* homolog, regulates AM formation and maintenance, and tillering is lost in the mutant [93]. The *BIF2* gene also positively regulates AM formation, and terminal meristem formation is impaired in the mutant [94]. Both are involved in auxin transport; *BIF2* is expressed upstream, and *BA1* is expressed downstream of auxin transport, integrating the genetic and hormonal control of AMs initiation [95]. Similarly, *BIF1* and *BIF4* are integral to the auxin signaling modules that regulate the boundary expression pattern of AM by dynamically regulating the expression of *BA1* [96]. Specifically, the dynamic efflux of auxin by the PIN (PIN-FORMED) protein family is essential for AM establishment [97]. PIN9 is a functional auxin efflux transporter in rice. The mutant *ospin9* has a low tiller number, and the *OsPIN9* overexpression strain has an increased tiller number [98]. Studies in *Arabidopsis* have demonstrated that *PIN3*, *PIN4,* and *PIN7* are likely important for communication between AMs and the main stem *PIN1*-dominated polar auxin transport stream [99]. *AtPIN3* is widely distributed, and its expression has been observed in root meristematic tissues as well as in the endodermal cells of young shoots [100]. Combined with previous studies and by comparing the results of differential enrichment of the transcriptome at nodes of different collateral types, we can speculate that at the 20C1734 node without axillary buds, auxin that accumulated from the beginning of leaf primordium development could not be properly exocytosed because of the silencing of genes of the PIN family of auxin exocytosis carriers, such as PIN3 and PIN4. This high concentration of auxin is the reason for the inability of AM at this location for region establishment, identity confirmation, and formation. In this process, ABA, as a plant hormone, also participated in this process. The common high content level of the two hormones led to the emergence of axillary bud-free nodes.

## 4. Methods and Materials

### 4.1. Plant Materials and Growth Conditions

The pepper (*Capsicum annuum*) normal lateral branches inbred line “20C1733” and less lateral branches “20C1734” was used as the material in this study, provided by College of Horticulture, China Agricultural University. 20C1734 is a mutant found in a long-term field experiment in our laboratory. After that, it was cultivated into an inbred line through continuous self-purification, and it has a stable character of few side branches, so it was used in this experiment. Unless stated otherwise, from seedling stage to flowering stage, pepper plants were placed in the growth room, and the environment was 27 °C for 16 h (full-spectrum light illumination) and 16 °C for 8 h (darkness). Water and pest control were performed according to standard protocol [101].

### 4.2. Cytological Experiment

The serial number of the axillary buds was not fixed because of their appearance time and unstable sequence. Therefore, the order of axillary buds was chosen to represent the number of leaf primordia in the true leaves. For example, the first real leaf was P_1_, and the second real leaf was P_2_. Here, we took the axillary bud at node P_6_ as the research object and grew in turn on the basis of P_6_ ≈ 1 cm in length. The axils of the leaves were cut and soaked in FAA fixative (5:6:89 ratio of formalin: glacial acetic acid:50% ethanol) [102]. The samples were embedded in paraffin, dehydrated in an ascending graded ethanol series, cleared with xylene for 30 min, and embedded in paraffin (Sigma, St. Louis, MO, USA) at 58 °C. The samples were cross-sectioned at 12 μm thickness cleared with xylene, rehydrated using a graded ethanol series, and stained with toluidine blue using standard protocols [103]. Slides were observed and photographed using an optical microscope (Olympus BX51, Olympus, Tokyo, Japan). All samples used to judge the growth cycle of Ams, the type of AM in the axils of different types of leaves were obtained from different plants, and all cytological sections were biological replicates performed in triplicate.

### 4.3. RNA-Seq Sample Preparation, Sequencing, and Analysis

The P_9_ leaf axils (with axillary buds) of 20C1733 as well as the P_3_ (with axillary buds) and P_9_ (without axillary buds) leaf axils of 20C1734 with similar growth status were analyzed at different developmental stages using RNA-seq. In addition, each leaf primordium was sampled at the same growth stage. Each plant was sampled only once, and tissues from 15 plants were combined as one sample. Each numbered sample had three biological replicates, recorded as “number 1, -2, and -3,” respectively. Leaf axillary positions were cut into 1 cm blade lengths with a V-shaped file in the growth chamber and then quickly placed into an RNA preservation solution (RNA-later, Life Technologies, Carlsbad, CA, USA) [104] and vacuum-treated (0.09 Mpa) for 30 min at 4 °C to ensure that the preservation solution penetrated into the cells.

All samples were modified under 12× magnification using a stereomicroscope (SMZ25, Nikon, Tokyo, Japan) to remove excess stem and leaf tissues to ensure the purity of the sample and the accuracy of RNA-Seq (Appendix A). Three biological replicates were analyzed for each sample. Total RNA was extracted from the modified sample using a quick RNA isolation kit (Huayueyang, Beijing, China) following the manufacturer’s instructions. The Illumina HiSeq 2000 platform (Illumina, San Diego, CA, USA) was used to generate 100-bp paired-end reads. Read counts per gene were calculated using fragments per kilobase of transcript per million mapped reads (FPKM), and numerous differentially expressed genes (DEGs) were obtained by comparing different tested samples. The genome compared in this step was the recently published pepper Ca_59 genome, which was used to locate the detected reads. Subsequent DEGs analysis, gene ontology (GO) annotation, and Kyoto Encyclopedia of Genes and Genomes (KEGG) enrichment analyses were performed as previously described [105]. The raw RNA-Seq sequence data were deposited in the Short Read Archive (SRA) of the National Center for Biotechnology (NCBI) and are available under accession number PRJNA922936. 

### 4.4. Expression Analysis Using Quantitative Real-Time PCR (RT-PCR)

cDNA was synthesized from 1 µg of RNA, which was obtained in 2.3 using a HiScript^®^ III 1st Strand cDNA Synthesis Kit (Vazyme, Nanjing, China). RT-PCR was performed on an ABI 7500 real-time PCR system (Applied Biosystems, Foster City, CA, USA) using the ChamQ Universal SYBR qPCR Master Mix (Vazyme, Nanjing, China) and the following cycling conditions: 95 °C for 30 s, and 40 cycles of 95 °C for 5 s and 60 °C for 30 s. Pepper *UBIQUITIN* (*Capana06g002873*) was used as the internal controls. Primers were designed online using the NCBI website (https://www.ncbi.nlm.nih.gov/tools/primerblast/, accessed on 19 February 2023); the information is listed in Appendix A. The 2^−∆∆Ct^ method was used to calculate the relative expression of each gene [106]. Three biological and three technical replicates (3 × 3) were performed for each gene.

### 4.5. Extraction and Quantification of Endogenous Indole-3-Acetic Acid (IAA), Trans-Zeatinriboside (TZR), Gibberellic Acid (GA_3_), and Abscisic Acid (ABA)

Approximately 0.1 g of samples harvested from the axillary buds of 20C1733 and 20C1734 were used to measure IAA, TZR, GA_3_, and ABA content, which reference the RNA-Seq sampling stage and processing method. Quantitative extraction was performed using ELISAs according to the method described by Liu et al. [107]. Three biological replicates were analyzed for each sample. In previous studies, this method has been used to measure endogenous plant hormones, such as auxin, zeatin, and gibberellin [108,109,110].

## 5. Conclusions

In this study, we found an inbred pepper line with reduced lateral branches, 20C1734, which had no axillary buds/lateral branches at the middle and upper nodes of the main stem. In pepper line 20C1733, which has an average number of lateral branches, successive sections of the leaf axils were used to confirm the different developmental periods of AM, thus confirming that 20C1734 without axillary buds at the nodes could not undergo normal AM region establishment and meristematic cell identity confirmation. Comparative transcriptomics of detailed samples revealed that hormone response and transport, especially the terms/pathways related to auxin and abscisic acid response, were enriched. Endogenous hormone assays revealed extremely high IAA and ABA levels in the leaf axils of axillary buds, whereas no regular changes in IAA levels were observed during AM formation. Accordingly, we speculate that the abnormal silencing of the PINs family genes, represented by *PIN3* and *PIN4*, in 20C1734 leads to the inability of the accumulated auxin to be transported away from the leaf axils of axil-less buds, which inhibits the confirmation of meristematic cell identity and regional establishment at the leaf axils, and with a high concentration of ABA accumulation, bud/lateral branch nodes without axils eventually form.

## Figures and Tables

**Figure 1 ijms-24-12718-f001:**
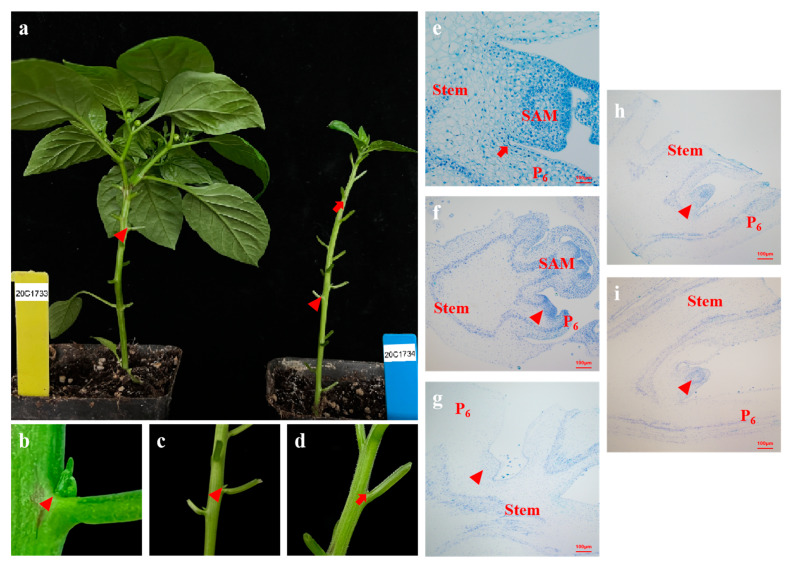
Phenotypes of less lateral branching plant 20C1734 and normal lateral branching plant 20C1733 and close-ups of each type of leaf axils and consecutive sections of different stages of AM development in the normal lateral branching parents. (**a**) Phenotypes of 20C1733 (left) and 20C1734 (right). (**b**) 20C1733 at node 9 (P_9_, with axillary buds); (**c**) 20C1734 at node 3 (P_3_, with axillary buds); (**d**) 20C1733 at node 9 (P_9_, without axillary buds). (**e**–**i**) 20C1733 paraffin section of P_6_ axil was observed when P_6_ (Primordium 6) < 1 cm, P_7_ < 1 cm, P_8_ < 1 cm, P_9_ < 1 cm, and P_10_ < 1 cm. Arrows in the figure indicate empty leaf axil sites, and triangles indicate areas that are establishing or AMs that have formed. SAM: Shoot Apical Meristem, P_6_: Primordium 6, Stem: Main stem.

**Figure 2 ijms-24-12718-f002:**
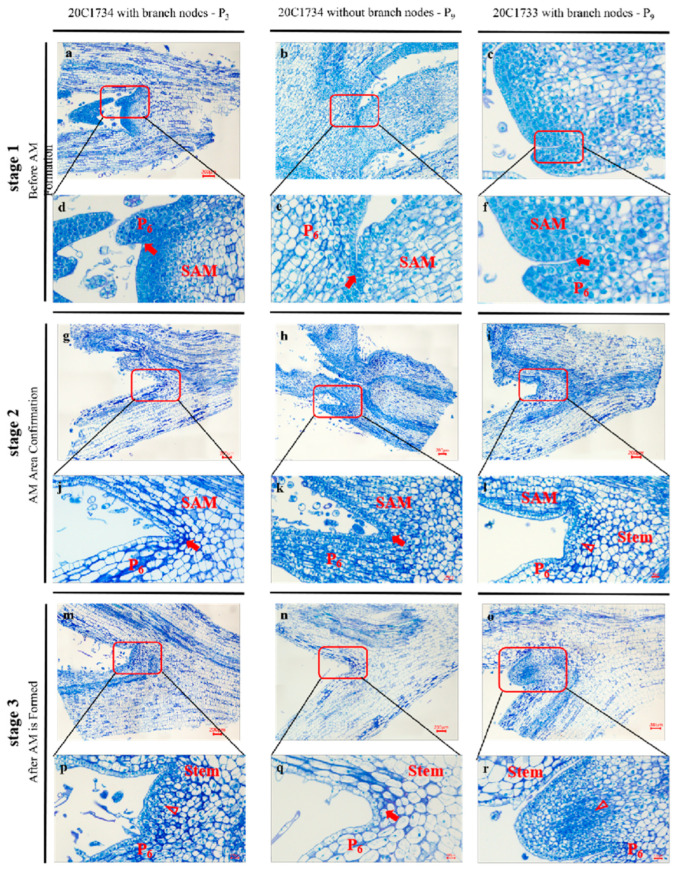
Cytological observations of different types of AM at different developmental stages. (**a**–**c**) Sections at the leaf axils of 20C1734 with axillary bud node P_3_, without axillary bud node P_9_ and 20C1733 with axillary bud node P_9_ before AM formation. (**d**–**f**) Enlarged close-up of the AM (to be determined) region corresponding to the (**a**–**c**) figures respectively. (**g**–**i**) Sections at the leaf axils of the same nodes during the establishment of the AM region. (**j**–**l**) Enlarged close-ups of the AM (to be determined) region corresponding to the (**g**–**i**) diagrams, respectively. (**m**–**o**) Sections at the leaf axils of the same nodes after the establishment of the AM region. (**p**–**r**) Enlarged close-ups of the AM (to be determined) region corresponding to the (**m**–**o**) diagram, respectively. Red arrows in the figure indicate empty leaf axil sites and red hollow triangles indicate areas being established or AMs that have formed. Bar = 200 μm.

**Figure 3 ijms-24-12718-f003:**
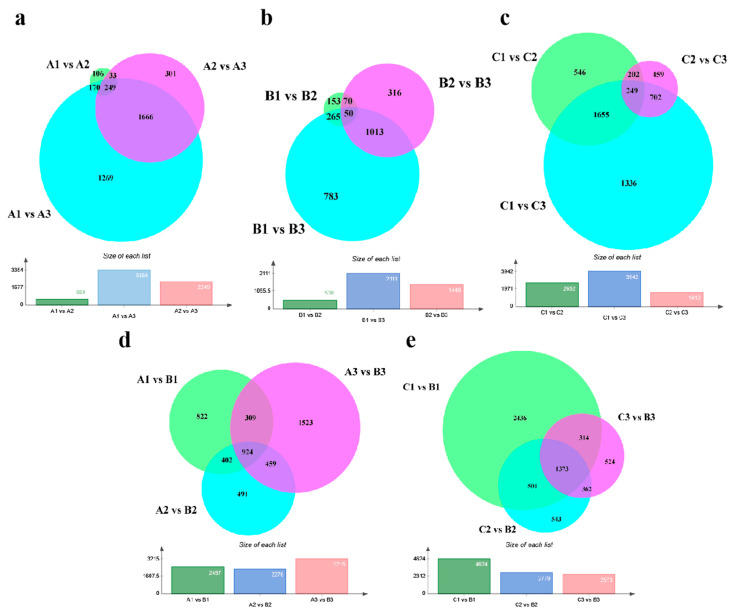
Comparison of the transcriptome relationships and DEGs of the examined samples. (**a**–**c**) Venn diagrams of DEGs obtained from position A, B, and C at different developmental stages. (**d**) Venn diagram of DEGs obtained by comparison between position C and B at the same stage. (**e**) Venn diagram of DEGs obtained by comparison between position A and B at the same stage. The bar below each Venn diagram indicates the number of DEGs accumulated between the comparison groups.

**Figure 4 ijms-24-12718-f004:**
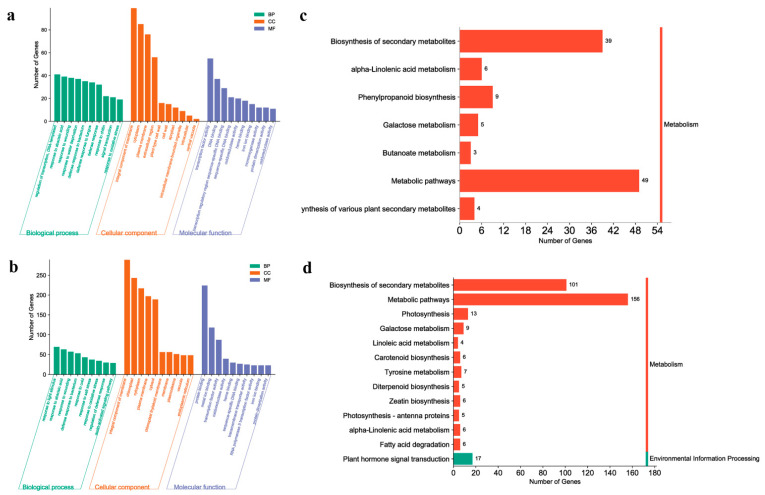
Comparative transcriptome analysis between B1 vs. B2 and C1 vs. C2. (**a**) GO enrichment analysis of B1 vs. B2. (**b**) GO enrichment analysis of C1 vs. C2. (**c**) KEGG enrichment pathway analysis of B1 vs. B2. (**d**) KEGG enrichment pathway analysis of C1 vs. C2. (**a**,**b**) Vertical coordinates indicate the number of enriched DEGs in each term, and horizontal coordinates indicate the names of the top 10 terms in each category in terms of the number of enriched DEGs. (**c**,**d**) Vertical coordinates indicate the names of the enriched pathways, and horizontal coordinates indicate the number of DEGs within each enriched pathway.

**Figure 5 ijms-24-12718-f005:**
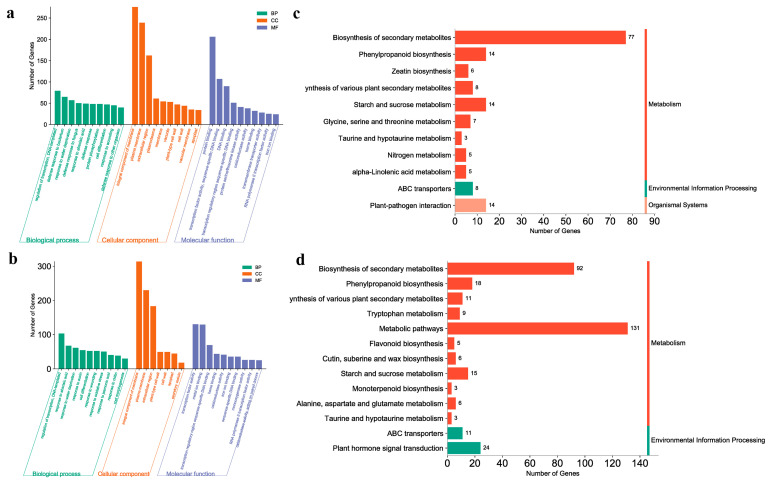
Comparative transcriptome analysis between B3 vs. B2 and C3 vs. C2. (**a**) GO enrichment analysis of B3 vs. B2. (**b**) GO enrichment analysis of C3 vs. C2. (**c**) KEGG enrichment pathway analysis of B3 vs. B2. (**d**) KEGG enrichment pathway analysis of C3 vs. C2. (**a**,**b**) Vertical coordinates indicate the number of enriched DEGs in each term, and horizontal coordinates indicate the names of the top 10 terms in each category in terms of the number of enriched DEGs. (**c**,**d**) Vertical coordinates indicate the names of the enriched pathways, and horizontal coordinates indicate the number of DEGs within each enriched pathway.

**Figure 6 ijms-24-12718-f006:**
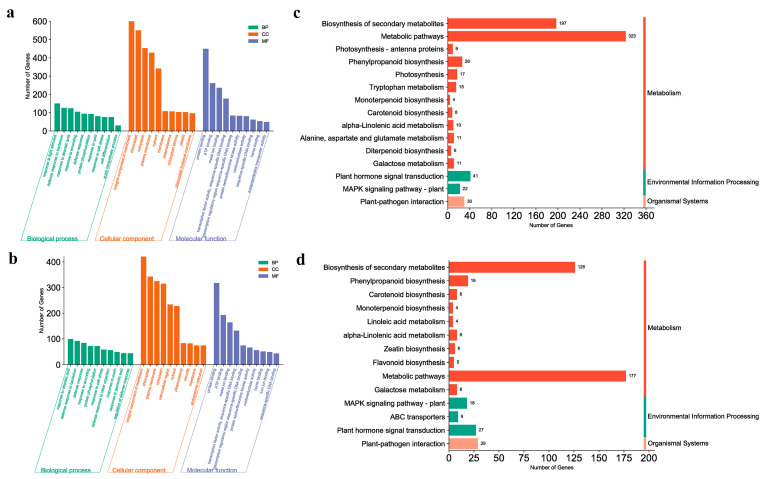
Comparative transcriptome analysis between C1 vs. B1 and C2 vs. B2. (**a**) GO enrichment analysis of C1 vs. B1. (**b**) GO enrichment analysis of C2 vs. B2. (**c**) KEGG enrichment pathway analysis of C1 vs. B1. (**d**) KEGG enrichment pathway analysis of C2 vs. B2. (**a**,**b**) Vertical coordinates indicate the number of enriched DEGs in each term, and horizontal coordinates indicate the names of the top 10 terms in each category in terms of the number of enriched DEGs. (**c**,**d**) Vertical coordinates indicate the names of the enriched pathways, and horizontal coordinates indicate the number of DEGs within each enriched pathway.

**Figure 7 ijms-24-12718-f007:**
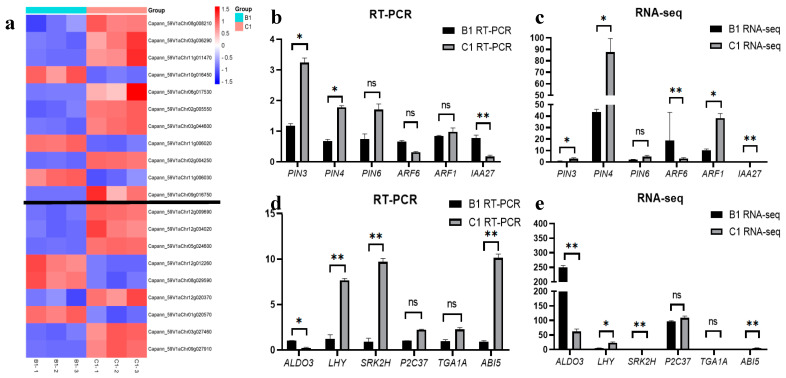
Heat map and RT−PCR validation of some genes related to potentially regulated enrichment pathway. (**a**) The expression heat map of some genes in the enrichment pathway in B1 and C1 detection groups, black lines are used to distinguish different terms/pathway. The horizontal axis represents each replicate of the comparison group, and the vertical axis indicates the gene number. (**b**) RT-PCR detection results of auxin-related genes. (**c**) Transcriptome sequencing results of auxin-related genes. (**d**) RT-PCR detection results of some genes in enrichment pathway. (**e**) Transcriptome sequencing results of some genes in enrichment pathway. All experiments were carried out three times in both technical repetition and biological repetition. Transcriptome determination data are plotted with CPM results obtained from analysis. “*” means significant difference (*p* < 0.05), “**” means significant difference (*p* < 0.01). ns stands for no significant difference.

**Figure 8 ijms-24-12718-f008:**
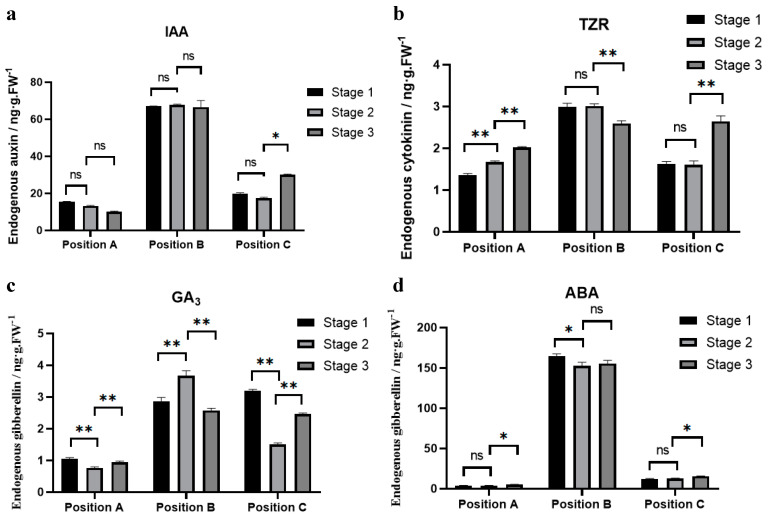
Determination of endogenous hormone content. (**a**–**d**) Content changes of IAA, TZR, GA_3_, and ABA in different parts at different developmental stages. The horizontal axis is different groups, and the vertical axis is hormone content. All experiments were carried out three times in both technical repetition and biological repetition. The significance analysis of content change among different parts is indicated by “*” and “**”. “*” means significant difference (*p* < 0.05), “**” means significant difference (*p* < 0.01), ns stands for no significant difference.

**Figure 9 ijms-24-12718-f009:**
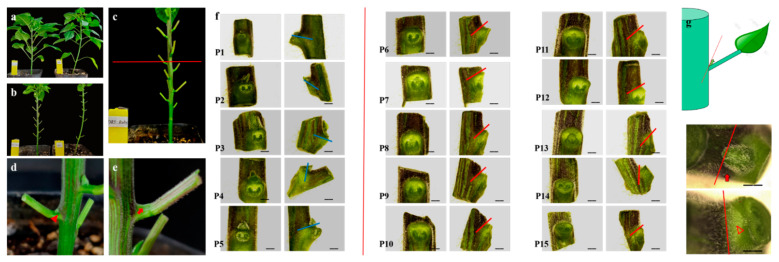
*DR5::Ruby* vacuumized and instantly infected the less lateral branch material 20C1734. (**a**) *DR5::Ruby* infested (left) versus non-infested (right) 20C1734 pepper. (**b**) Plants with leaves removed. (**c**) *DR5::Ruby* infested plants, there are no axillary buds at each of the leaf axils above the red line, and the leaf axils with axillary buds are below. (**d**,**e**) *DR5::Ruby* with axillary buds of leaf axils (**d**) and without axillary buds of leaf axils (**e**). (**f**) Front and side views of each leaf axils of *DR5::Ruby* plant under stereomicroscope. The left side of the red line shows the leaf axils with axillary buds and the right side shows the leaf axils without axillary buds. The blue line and the red line have the boundaries of auxin distribution at lateral branches and non-lateral branches, respectively. (**g**) Schematic diagram of leaf axils, with the crease where the stem is attached to the leaf, the leaf axils with axillary buds on the upper right, and the leaf axils without axillary buds on the lower right. The red line in the figure indicates the distribution boundary of *DR5::Ruby* in the axils of leaves. The bar in (**f**) = 5 mm and the bar in (**g**) = 2 mm. Arrows in the figure indicate empty leaf axil sites, and triangles indicate areas that are establishing or AMs that have formed.

**Table 1 ijms-24-12718-t001:** Description of time and location of transcriptome sampling.

	Position A	Position B	Position C
Stage 1	20C1734 node P_3_ with axillary buds, before AM formation (A1)	20C1734 node P_9_ without axillary buds, before AM formation (B1)	20C1733 node P_9_ with axillary buds, before AM formation (C1)
Stage 2	20C1734 node P_3_ with axillary buds, AM area confirmation (A2)	20C1734 node P_9_ without axillary buds, AM area confirmation (B2)	20C1733 node P_9_ with axillary buds, AM area confirmation (C2)
Stage 3	20C1734 node P_3_ with axillary buds, after AM is formed (A3)	20C1734 node P_9_ without axillary buds, after AM is formed (B3)	20C1733 node P_9_ with axillary buds, after AM is formed (C3)

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
