# Peer review of "Characterization of the Molecular Events Underlying the Establishment of Axillary Meristem Region in Pepper"

_ijms, 2023, doi:10.3390/ijms241612718_

Round 1

Reviewer 1 Report

In this study, the authors investigate the role of shoot branching patterns in determining the aerial architecture of plants, emphasizing plant diversity as a major factor. Specifically, they focus on an inbred pepper line, 20C1734, characterized by reduced lateral branches. This line exhibited an absence of lateral branches at the middle and upper nodes of the main stem, resulting in smooth and flat leaf axils. The researchers aimed to elucidate the underlying molecular and physiological mechanisms governing this unique branching pattern. 

This study provides valuable molecular insights into the shoot branching patterns and axillary meristem development in the 20C1734 inbred pepper line. The authors successfully identified critical periods of axillary meristem establishment, differentially expressed genes related to endogenous phytohormones, and the aberrant hormone profiles in leaf axils lacking axillary buds. These findings contribute to our understanding of plant architecture and highlight the importance of hormonal regulation in determining branching patterns. The research opens up possibilities for further investigations aimed at manipulating plant architecture to enhance crop productivity and quality.

As a conclusion, the paper would be sufficient to merit publication in IJMS though a revision is recommended which needs to include the following points.

(1) The notation of samples for RNA-seq analysis is position A, B, and C, but it is difficult to understand. I think it should be changed to specific names.

(2) The detail about the screening of 20C1734 should be described in material and methods or results sections. 

(3) Figure 3. If the number of genes is shown in the bar chart, I think the size of the circle in the Venn diagram should be proportional to the number of genes. 

(4) Section 3.7 (Figure 9). The explanation is difficult to understand and needs to be completely rewritten so that the experimental principles, etc. can be understood.

(4) Figure 9. panel f is too small.

(5) Figure 9. panel d and f. It would be easier to understand if you put arrows on the important parts.

Author Response

Response to Reviewer 1 Comments

General comments: In this study, the authors investigate the role of shoot branching patterns in determining the aerial architecture of plants, emphasizing plant diversity as a major factor. Specifically, they focus on an inbred pepper line, 20C1734, characterized by reduced lateral branches. This line exhibited an absence of lateral branches at the middle and upper nodes of the main stem, resulting in smooth and flat leaf axils. The researchers aimed to elucidate the underlying molecular and physiological mechanisms governing this unique branching pattern. 

This study provides valuable molecular insights into the shoot branching patterns and axillary meristem development in the 20C1734 inbred pepper line. The authors successfully identified critical periods of axillary meristem establishment, differentially expressed genes related to endogenous phytohormones, and the aberrant hormone profiles in leaf axils lacking axillary buds. These findings contribute to our understanding of plant architecture and highlight the importance of hormonal regulation in determining branching patterns. The research opens up possibilities for further investigations aimed at manipulating plant architecture to enhance crop productivity and quality.

As a conclusion, the paper would be sufficient to merit publication in IJMS though a revision is recommended which needs to include the following points.

First of all, thank you very much for your approval and patient revision of this article. We have gained a lot from your comments. The replies to the comments are listed below, and the corresponding parts of the manuscript have also been modified according to your comments. Thank you again for your painstaking effort on this article!

  • The notation of samples for RNA-seq analysis is position A, B, and C, but it is difficult to understand. I think it should be changed to specific names.

We had asked the same question to ourself. On this point, we first used this expression in the writing of the article, but because the similar expression of "20C1733- with side branches and nodes" is too cumbersome and cumbersome, we adopted the ways of A, B and C to refer to it. In order to better understand, on the one hand, we draw a table in the text, on the other hand, we draw a schematic diagram in the form of attached drawings for easy understanding.

  • The detail about the screening of 20C1734 should be described in material and methods or results sections.

This part has been added to Part 2.1 of Material and methods. Thank you for your suggestion. It was our negligence and we are very sorry.

  • Figure 3. If the number of genes is shown in the bar chart, I think the size of the circle in the Venn diagram should be proportional to the number of genes.

Thank you for your correct suggestion, which we didn't consider in the previous experiment. According to your suggestion, I redrawn Venn diagram, and drew a circle with response scale according to the number. Thank you for your help!

4) Section 3.7 (Figure 9). The explanation is difficult to understand and needs to be completely rewritten so that the experimental principles, etc. can be understood.

Thank you for your suggestion, and I am very sorry for the trouble caused to your review due to the reasons we expressed. The ambiguous sentences in this paragraph have been modified according to your requirements, and it should be easy to understand now.

5) Figure 9. panel f is too small.

I'm very sorry that this was an oversight in our drawing process. According to your suggestion, I have revised Figure 9 as a whole now, and I hope this result will satisfy you.

  • Figure 9. panel d and f. It would be easier to understand if you put arrows on the important parts.

According to your suggestion, the corresponding positions (D, E, F) in Figure 9 have been marked accordingly, and now the picture looks more perfect. Thank you for your advice, which makes the article look more full and up to standard..

Finally, on behalf of all the authors, please allow me to express my sincere thanks for your review of this article and your suggestions! Good luck in your work!

Reviewer 2 Report

your paper describes the characterization of a reduced branching capsicum mutant. It provides some interesting insights into AM formation.

Its interesting because modern approaches would go about mapping the mutation and then characterizing it from there. Your approach is very much 'old school', but equally valid, (not criticizing here) whereby you've characterized what processes the mutation(s) affect without identifying it.

The paper is mostly fine apart from some minor English edits and I recommend re-writing one paragraph in the introduction where you allude to the strigolactone side of branching. My main criticism is the quality of the figures. These need a major re-work because they are illegible even at 150% zoom. The font sizes are too small and some of the graphs are too low definition. You will have to be imaginative and present it in a different way. see attached pdf for comments

generally fine but a few edits required, see previous.

Author Response

Response to Reviewer 2 Comments

your paper describes the characterization of a reduced branching capsicum mutant. It provides some interesting insights into AM formation.

It is interesting because modern approaches would go about mapping the mutation and then characterizing it from there. Your approach is very much 'old school', but equally valid, (not criticizing here) whereby you've characterized what processes the mutation(s) affect without identifying it.

The paper is mostly fine apart from some minor English edits and I recommend re-writing one paragraph in the introduction where you allude to the strigolactone side of branching. My main criticism is the quality of the figures. These need a major re-work because they are illegible even at 150% zoom. The font sizes are too small and some of the graphs are too low definition. You will have to be imaginative and present it differently. see attached PDF for comments. 

Firstly, thank you very much for reviewing this article after work, which is very important for our work. Thank you for your affirmation of this article and your questions. 

Regarding the question about the text in the article you raised, it has been revised according to your comments on the article. Thank you very much for reading the content of the article in detail and putting forward very constructive suggestions.

At the same time, we are very sorry that the resolution of most of the pictures in your article does not meet the requirements. This point was ignored in the writing process of the article, which is very inappropriate, and we are very sorry for this. Now, in the revised version of the article, Figures 3- 9 have been redrawn to ensure high resolution and clarity, and now they meet the publishing standards. I'm terribly sorry.

Finally, on behalf of all the authors, please allow me to express my sincere thanks for your review of this article and your suggestions! Good luck in your work!